# Evaluation of Different Types of Stimuli in an Event-Related Potential-Based Brain–Computer Interface Speller under Rapid Serial Visual Presentation

**DOI:** 10.3390/s24113315

**Published:** 2024-05-22

**Authors:** Ricardo Ron-Angevin, Álvaro Fernández-Rodríguez, Francisco Velasco-Álvarez, Véronique Lespinet-Najib, Jean-Marc André

**Affiliations:** 1Departamento de Tecnología Electrónica, Instituto Universitario de Investigación en Telecomunicación de la Universidad de Málaga (TELMA), Universidad de Málaga, 29071 Malaga, Spain; afernandezrguez@uma.es (Á.F.-R.); fvelasco@uma.es (F.V.-Á.); 2Laboratoire IMS, CNRS UMR 5218, Cognitive Team, Bordeaux INP-ENSC, 33400 Bordeaux, France; veronique.lespinet@ensc.fr (V.L.-N.); jean-marc.andre@ensc.fr (J.-M.A.)

**Keywords:** brain-computer interface (BCI), event-related potential (ERP), rapid visual serial presentation (RSVP), stimulus, speller

## Abstract

Rapid serial visual presentation (RSVP) is currently a suitable gaze-independent paradigm for controlling visual brain–computer interfaces (BCIs) based on event-related potentials (ERPs), especially for users with limited eye movement control. However, unlike gaze-dependent paradigms, gaze-independent ones have received less attention concerning the specific choice of visual stimuli that are used. In gaze-dependent BCIs, images of faces—particularly those tinted red—have been shown to be effective stimuli. This study aims to evaluate whether the colour of faces used as visual stimuli influences ERP-BCI performance under RSVP. Fifteen participants tested four conditions that varied only in the visual stimulus used: grey letters (GL), red famous faces with letters (RFF), green famous faces with letters (GFF), and blue famous faces with letters (BFF). The results indicated significant accuracy differences only between the GL and GFF conditions, unlike prior gaze-dependent studies. Additionally, GL achieved higher comfort ratings compared with other face-related conditions. This study highlights that the choice of stimulus type impacts both performance and user comfort, suggesting implications for future ERP-BCI designs for users requiring gaze-independent systems.

## 1. Introduction

A brain–computer interface (BCI) grounded in event-related potentials (ERPs) can serve as assistive technology (AT), empowering individuals to interact with their surroundings solely through brain signals [1]. Beyond ERP-BCIs, various other ATs exist for communication purpose, encompassing eye-tracking systems, head-pointing devices, and low-pressure sensors [2]. Nevertheless, specific injuries or ailments, such as amyotrophic lateral sclerosis (ALS), can compromise muscular function and even eye movements, rendering some of these ATs less effective [3]. Consequently, in cases of profound motor limitations, many conventional AT solutions may lose their utility due to their reliance on muscular pathways that could be impaired in patients [4,5]. This underscores the potential of ERP-BCIs as a promising alternative in severe instances of muscular control impairment. In addition to other applications such as home appliance control [6], one of the most developed applications for these patients is spellers, systems that enable verbal communication through letter selection for constructing words and sentences [7].

ERPs manifest as alterations in the brain’s electrical activity voltage are induced by the perception of specific events. These events encompass external stimuli delivered through various sensory channels, including visual, auditory, or tactile stimuli [8]. For the purposes of this study, we have opted for the visual modality. As outlined by Allison et al. [8], this modality typically yields superior outcomes in the context of ERP-BCI control. Furthermore, the visual modality can be effectively employed in specific presentation paradigms, even when the user lacks control over their gaze. One such paradigm that does not necessitate ocular mobility is the rapid serial visual presentation (RSVP) [9]. Below, we elucidate the utilisation of RSVP for controlling a visual ERP-BCI.

The key characteristic of RSVP lies in its sequential display of visual stimuli, presented one after another, all within the same spatial location. In the context of operating a visual ERP-BCI, various visual stimuli are shown to the user, who must focus on a specific one. Directing attention to the intended stimulus, such as a letter in a spelling application, elicits a distinct brain electrical signal compared to signals associated with unintended stimuli. Therefore, the primary objective of an ERP-BCI is to differentiate between the desired or attended stimulus (referred to as the target) and the undesired or non-attended stimuli (known as non-targets) based on the user’s brain activity. The key component utilized by these systems is the P3 signal, also known as P300. This signal signifies a positive peak in the brain’s electrical activity, typically occurring around 300–600 ms following the presentation of an expected stimulus to the user [10]. However, ERP-BCI applications commonly incorporate a broader range of ERPs within this timeframe (e.g., P200, N200, or a late positive potential). Essentially, any signal aiding in distinguishing the attended stimulus (target) from the unattended ones (non-targets) will be integrated within the specified time interval (e.g., 0–800 ms after stimulus onset) [11,12].

As highlighted previously, the target audience for a visual ERP-BCI may include individuals who have lost the ability to control their eye movements. Consequently, it is crucial to customize the interfaces provided to these users according to their capabilities. Importantly, performance can significantly decline when users are unable to shift their gaze towards stimuli [13,14]. Therefore, employing paradigms that do not rely on eye control to achieve satisfactory performance, such as the RSVP paradigm, can be advantageous. Additionally, prior studies have shown that various factors, such as (i) the spatial arrangement of stimuli [15], (ii) the duration of stimulus presentation [16], or (iii) the selection of stimulus type [17], influence performance levels.

The selection of stimuli in an ERP-BCI has been extensively investigated, particularly in gaze-dependent paradigms such as matrix-based ones where stimuli are dispersed across different locations within the matrix. One of the most commonly used matrix-based paradigms is the row-column paradigm (RCP) [18]. In the RCP, rows and columns are highlighted sequentially from grey to white. To choose a character, the user focuses on the flashing of a specific target character, which acts as the task-relevant stimulus that triggers the ERP component, such as P300. Once the ERP is associated with a particular row and column, the BCI can infer the user’s intended character. In these matrix-based paradigms, images of faces have consistently proven to be highly effective stimuli [19]. Building on this trend, recent research has explored how even the colour of these faces can impact performance. Notably, Li et al. [20] demonstrated that semi-transparent green faces outperformed regular-coloured, semi-transparent faces. Subsequently, Li et al. [21] investigated the influence of using semi-transparent faces of different colours—blue, green, and red—superimposed on the letters. That study revealed that red faces yielded better results compared with green and blue faces. However, it is crucial to recognise that the performance outcomes and stimulus preferences observed in matrix-based paradigms may not necessarily hold true in the RSVP paradigm [22]. Consequently, it becomes intriguing to investigate whether the observed impact of face colour on performance can be replicated under the RSVP modality. Therefore, the objective of this study is to reproduce the experiment proposed by Li et al. [21]—which, to our knowledge, presents the latest advance in terms of stimuli with better performance in a matrix-based paradigm—but within the context of the RSVP paradigm.

In conclusion, RSVP stands out as a viable gaze-independent control paradigm employed in the field of BCIs, particularly when users lack oculomotor control. Nevertheless, the impact observed in previous studies regarding the colour of faces used as visual stimuli, which primarily pertains to gaze-dependent paradigms, remains unexplored within the context of RSVP. Therefore, investigating the influence of stimulus type on performance in an ERP-BCI under RSVP holds the potential to make a noteworthy contribution to the field.

## 2. Methods

### 2.1. Participants

This study included 15 healthy French-speaking participants (aged 26.87 ± 11.32, 10 women and 5 men, identified as P01 to P15). Each participant possessed either normal vision or vision corrected to normal, and they all gave written consent. Approval for the study was obtained from the Ethics Committee of the University of Malaga, and it adhered strictly to ethical guidelines. None of the individuals had a prior history of neurological or psychiatric disorders, nor were they taking any medication that could potentially impact the experiment. Prior to commencing the study, all participants were briefed on the experimental protocol and were given the freedom to withdraw from participation at any point.

### 2.2. Data Acquisition and Signal Processing

The electroencephalographic (EEG) data were recorded using specific electrode positions based on the 10/10 international system, including Fz, Cz, Pz, Oz, P3, P4, PO7, and PO8. These channels were referenced to the right earlobe, while FPz served as the ground. Signal amplification was conducted using a 16-channel gUSBamp amplifier (Guger Technologies GmbH, Schiedlberg, Austria). The amplifier was configured with a bandpass filter spanning 0.1 to 60 Hz, a 60 Hz notch filter activated, and a sensitivity set to 500 μV. EEG signals were digitized at a sampling rate of 256 Hz. To manage EEG data acquisition and processing, this study employed the UMA-BCI Speller software (v0.45) [23], an open-source BCI speller application developed by the UMA-BCI group (https://umabci.uma.es). This software is built upon the widely recognised BCI2000 platform (v3.0.5) [24], ensuring reliability. The UMA-BCI Speller simplifies the configuration and usage of BCI2000, providing a more user-friendly interface. This study utilized a stepwise linear discriminant analysis (SWLDA) approach on EEG data, akin to the methodology employed in developing a BCI speller using BCI2000. The SWLDA facilitated feature extraction, classifier weight determination, and accuracy assessment. In this context, features represent signals at specific EEG channels and time points following the stimulus. A comprehensive description of the SWLDA algorithm is available in the P300Classifier user reference [25], where it is outlined as a process “to derive a final linear model that approximately fits a dataset (stimulus) by employing multiple linear regressions and iterative statistical methods, thereby selecting only significant variables for inclusion in the final regression”. The default configuration was applied, setting the maximum number of features to 60 and maximum *p*-values for feature inclusion or exclusion at 0.1 and 0.15, respectively. The default time interval analyzed was 0–800 ms post-stimulus presentation. Subsequently, subject-specific weights for the classifier were obtained through this analysis, and the classifier was then applied to the EEG data to determine the item each subject attended to.

### 2.3. Experimental Conditions

In this study, four different RSVP paradigms have been assessed, each distinguished by the type of stimulus employed in accordance with those used by Li et al. [21]: (i) grey letters (GL), (ii) semi-transparent grey letters with a red famous face (RFF), (iii) semi-transparent grey letters with a green famous face (GFF), and (iv) semi-transparent grey letters with a blue famous face (BFF) (Figure 1). Each paradigm displayed six distinct letters (A, E, I, N, R, and S, in Arial font), which were utilised for word formation during the experiment. This specific number of letters was chosen to ensure that the target selection time remained manageable, as the primary goal was to validate these diverse sets of stimuli for communication purposes within the RSVP framework. These letters were deliberately selected to facilitate both calibration and real-time writing tasks, aligning with prior studies that employed a similar number of elements to test hypotheses [17,22]. As a stimulus, the famous face of David Beckham was utilised, a choice consistent with previous studies [20]. The dimensions of the stimuli were standardised, with letters measuring approximately 6 × 6 cm (using the letter “N” as a reference) and faces spanning around 6 × 8 cm. The interface background was set to black, and the stimuli were presented centrally on the screen. Additionally, at the top of the screen, both the letters available for selection and those already chosen were indicated. Each stimulus was centrally displayed on the screen for 187.5 ms, with an inter-stimuli interval (ISI) of 93.75 ms, resulting in a stimulus onset asynchrony (SOA) lasting 281.25 ms. The time taken to complete a sequence (comprising the presentation of each stimulus) was 1687.5 ms. Each trial consisted of several sequences (fixed at 10 in the calibration task and variable in the online task, as explained in Section 2.4). Therefore, the flashing of stimuli lasted 1687.5 ms multiplied by the number of sequences used. Before starting a trial, there was a pause of 4000 ms, as well as another pause of 875 ms at the end of it. Thus, between one selection and the commencement of the next (i.e., between completed sets of sequences) there was a total pause of 4875 ms.

### 2.4. Procedure

A within-subject design was used, ensuring that all participants experienced all experimental conditions (Figure 2). The entire session lasted approximately 90 min. To prevent potential biases like learning or fatigue, the order of the paradigms was counterbalanced among the participants. Each condition comprised two BCI tasks: (i) an initial calibration phase aimed at capturing the user-specific signal patterns, during which no feedback was provided, and (ii) an online phase, where participants actively controlled the interface. Both phases involved the task of composing four-letter French words. Finally, in a subjective questionnaire phase, the participants were required to respond to items related to their opinion regarding the control experience during the just-completed condition. In the calibration phase, the participants wrote four words (“ASIE”, “REIN”, “NIER”, and “SAIN”), totalling 16 letter selections. In the online phase, participants were required to write three different words chosen freely (although the following four were suggested: “ANIS”, “AIRE”, “REIN”, and “SERA”), resulting in 12 letter selections. The participant had to indicate before starting the writing which word they wanted to spell out. In the event of an erroneous letter selection during the online phase, the participant proceeded to the next letter. There were short breaks between words in both phases, with the duration varying based on the participant’s preference. During the calibration phase, the number of sequences, representing how frequently each stimulus was presented, was fixed at 10. Conversely, for the online phase, the number of sequences employed for character selection corresponded to the instance in which the participant achieved the second-highest consecutive accuracy results in the calibration task. In instances where the maximum accuracy was not repeated consecutively or only occurred once, the first-best sequence was chosen.

### 2.5. Evaluation

Several variables were analyzed to assess how the choice of stimulus in each experimental condition influenced letter selection performance, subjective user experience, and the EEG signal’s ERP waveform.

#### 2.5.1. Performance

To assess the impact of the RSVP paradigm and stimulus type on online performance, two parameters were utilised: (i) accuracy (%), which represents the proportion of correctly selected letters; and (ii) the information transfer rate (ITR, bit/min), which aims to assess the communication speed of the system (e.g., Li et al. [21]). The ITR considers accuracy, the number of elements available on the interface, and the time required to select one element:$$ITR=\frac{\log_{2}N+P\log_{2}P+\left(1-P\right)\log_{2}\frac{1-P}{N-1}}{T},$$
where *P* represents the system’s accuracy, *N* denotes the number of elements accessible in the interface, and *T* signifies the time required to complete a trial (i.e., select an element). The ITR calculation did not account for the pauses between selections.

#### 2.5.2. ERP Waveform

To examine the impact of different experimental conditions on ERPs, the relative amplitude of target and non-target stimuli as well as the amplitude difference between them during the calibration phase were explored. This amplitude difference, previously utilised as a metric [20,26], provides valuable insights into the ERP paradigm, especially when a target stimulus competes with non-target stimuli. Thus, to study the different effects of target and non-target stimuli, measuring the amplitude difference is considered more convenient than directly assessing the target stimulus’s amplitude. The evaluation covered a time interval spanning −200 to 1000 ms, with a baseline period from −200 to 0 ms. Data artefacts were rectified by using the artifact subspace reconstruction (ASR) algorithm with default settings in EEGLAB (v2022.1), coupled with the Riemannian distance [27]. The ASR is a “non-stationary method based on a PCA [principal component analysis] window, dedicated to automatically detecting and removing artifacts” [28]. Additionally, a low-pass filter at 30 Hz was applied.

#### 2.5.3. Subjective Items

Three specific variables related to the user’s perception of the control experience were collected at specific points during the experiment. First, the participants rated their perceived level of fatigue on a scale from 0 (no fatigue) to 5 (high fatigue) after using each condition. Second, the participants rated letter visibility on a scale from 0 (not visible at all) to 5 (completely visible). Third, the participants ordered the conditions based on their level of comfort experienced, with this ranking completed at the end of the session after experiencing all conditions. The scoring for comfort ranged from 1 to 4 (the number of experimental conditions), with higher scores indicating greater comfort with the condition.

#### 2.5.4. Statistical Analyses

We used the tidyverse package [29] in R [30] to examine how stimulus type affected each variable related to performance or subjective items. For each variable, we initially conducted an analysis encompassing all conditions involved. If significant differences were found, pairwise comparisons between specific conditions were performed. Parametric methods were employed for statistical analysis when the variables’ distributions met the normality criterion: analysis of variance (ANOVA) for comparing means of three or more conditions and paired *t*-tests for comparing means of two specific conditions. However, when the normality assumption was violated, non-parametric methods were employed: the Friedman test as an alternative to ANOVA and the Wilcoxon signed-rank test as an alternative to the Student’s *t*-test. To mitigate the risk of false positives (i.e., rejecting the null hypothesis when it is true) arising from multiple comparisons, we applied the Bonferroni correction method.

Additionally, for the analysis of the ERP waveform, we utilized the EEGLAB software (v2022.1) [31] to conduct permutation-based (non-parametric) statistics. This involved comparing the amplitudes of target and non-target stimuli, as well as the amplitude difference between them, across all channels for each paradigm. To account for multiple comparisons across channels and intervals simultaneously, these analyses were corrected using the false-discovery rate (FDR) method [32].

## 3. Results

### 3.1. Performance

Table 1 presents the number of sequences utilised, the accuracy, and the ITR for each participant and condition during the online task. The average number of sequences employed was as follows: GL, 4.87 sequences; RFF, 5.33 sequences; GFF, 5.53 sequences; and BFF, 4.73 sequences. The average accuracy for each condition was as follows: GL, 83.3%; RF, 87.2%; GFF, 96.1%; and BFF, 88.3%. Lastly, the average ITR for each condition was as follows: GL, 14.2 bit/min; RFF, 14.2 bit/min; GFF, 17.7 bit/min; and BFF, 16.1 bit/min. The statistical analysis revealed significant differences in terms of accuracy between conditions (*χ*^2^(3) = 12.4; *p* = 0.006). Specifically, the GFF condition exhibited higher accuracy than the GL condition (*p* = 0.032). However, neither the number of sequences (*χ*^2^(3) = 0.508; *p* = 0.917) nor the ITR (*F*(3, 42) = 1.816; *p* = 0.159) differed significantly based on the type of stimulus used.

### 3.2. ERP Waveform

Figure 3 presents the signal related to target and non-target stimuli as well as the amplitude difference between the two types of stimuli. Both target and non-target stimuli show a consistent visual evoked potential (VEP) approximately every 285 ms, which coincides with the SOA applied in the experiment and is particularly pronounced in channels related to occipital regions (PO7, PO8, and Oz). Therefore, it can be stated that this VEP corresponds to the mere presentation of each stimulus in the interface; the signal of the target stimulus is also affected by this VEP because a target stimulus is temporally surrounded by non-target stimuli. However, for target stimuli, there is a particularly pronounced potential around 500 ms, which could be interpreted as P300. This interpretation is further supported when observing the signal related to the amplitude difference, where the effect of VEPs is nullified. Therefore, this potential could be crucial in assisting the classifier in discriminating between different types of stimuli to determine to which the user is attending. For the amplitude difference, we observed a negative component around 350–450 ms in occipital channels, which could be interpreted as a potential N200 component [33]. It is also worth noting that both N200 and P300 exhibited a shorter latency in the GL condition, which resulted in significant differences in the plot related to the amplitude difference around 350–550 ms for most of the recorded channels.

Additionally, various analyses have been conducted, and their corresponding figures have been added as Appendix A. Firstly, an analysis was performed to compare the target signal of each stimulus against the non-target signal (Appendix A). This analysis corroborates the observations made through the amplitude difference graphs in Figure 3: the segments with the most significant differences between target and non-target signals occurred during the time interval associated with the P300 component across all recorded channels, as well as a possible N200 component for channels located in the occipital region (PO7, PO8, and Oz). Furthermore, for the amplitude difference, we conducted multiple comparison analyses between each pair of conditions: GL versus RFF (Appendix A), GL versus GFF (Appendix A), GL versus BFF (Appendix A), RFF versus GFF (Appendix A), RFF versus BFF (Appendix A), and GFF versus BFF (Appendix A). These analyses revealed that the greatest difference in amplitude between condition pairs was observed between GL versus RFF and GFF. These findings suggest that latency in the letter condition is shorter for both the P300 and N200 components.

### 3.3. Subjective Items

Table 2 displays the results concerning subjective items related to fatigue, letter visibility, and comfort. The average fatigue for each condition was GL, 2.47 points; RF, 2.93 points; GFF, 2.8 points; and BF, 3 points. The average visibility of stimuli for each condition was GL, 3.67 points; RFF, 2.93 points; GFF, 2.8 points; and BFF, 2.6 points. Lastly, the comfort level for each condition was GL, 3.67 points; RFF, 2 points; GFF, 2 points; and BFF, 2.33 points. First, the analyses did not reveal significant differences in fatigue (*F*(3, 42) = 2.031; *p* = 0.124). This implies that the type of stimulus utilized does not conclusively affect the level of fatigue experienced by participants. Second, although the employed analysis indicated significant differences among the conditions regarding visibility (*χ*^2^(3) = 9.02; *p* = 0.029), post hoc multiple comparisons did not reveal any significant findings. Finally, the analysis concerning the comfort variable showed a significant influence of the type of stimulus used (*χ*^2^(3) = 17; *p* < 0.001). Thus, in this context, it appears that the type of stimulus employed indeed influenced the participant comfort levels during system control. Specifically, there were significant differences between the GL condition and those with faces (RFF, *p* = 0.045; GFF, *p* = 0.017; BFF, *p* = 0.028). Therefore, it appears that the use of faces decreases the comfort level during the interface usage.

## 4. Discussion

The aim of this work was to explore the effect of face colour—previously studied in matrix-based paradigms—on an ERP-based BCI under RSVP. We analyzed three dimensions: (i) performance, (ii) ERP waveform, and (iii) subjective items. In this section, we discuss the results for each dimension in the context of the previous literature.

### 4.1. Performance

The average performance of the conditions was >80% accuracy and >14 bit/min, which reflects quite positive results in the context of gaze-independent ERP-BCI spellers [34]. In fact, the GFF condition achieved an accuracy of 96.1% and an ITR of 17.7 bit/min. It is challenging to contextualise the results obtained in the present study with those from the previous literature due to the use of different paradigms or experimental specifications (e.g., SOA). However, Ron-Angevin et al. [35] used the same configuration to explore the use of three types of stimuli under RSVP: white letters, natural-coloured famous faces without a letter, and pictures without a letter. The results of that study were similar to our findings (Table 3). The GFF condition of the present study and the famous face condition of Ron-Angevin et al. [35] stand out in terms of the ITR: they were the only conditions with an accuracy > 90% and an ITR > 17 bit/min.

Regarding the hypothesis proposed in the present study, there was a tendency for the GFF condition to exhibit higher performance, even demonstrating significantly superior accuracy compared with the GL condition. Previous studies that involved matrix-based paradigms have found that faces performed better than letters [19] and that there were even differences between faces based on their colour [20,21]. Specifically, Li et al. [21] showed that red faces performed better than green ones. Therefore, our results are only partially aligned with these previous works since differences were observed only between green faces (GFF) and letters (GL) but not between other coloured faces (red or blue) and letters nor between any other face colour. It should also be noted that the significant differences observed in performance were only with respect to accuracy, not the ITR. Therefore, our results should be interpreted with caution, and it is advisable that future studies further validate these findings. It is important to remember that other studies conducted using RSVP have shown that the choice of stimulus does not necessarily result in enhanced performance compared with what is achieved in other matrix-based paradigms [22,35]. Hence, it is crucial to consider the specific characteristics of each paradigm to identify the variables that can contribute to enhance performance. Furthermore, the possibility of false positives in the statistical analyses should not be overlooked, both in the present study and in previous ones. Therefore, it is of the utmost importance to emphasise the need for experimental replication.

Our proposal has employed a gaze-independent paradigm, the RSVP, which has demonstrated promising results. However, other paradigms and modalities may also be potentially suitable for patients without oculomotor control. Some of these visual paradigms include those previously demonstrated by, for example, [36] or [37], which are based on the use of covert attention. While these systems perform adequately, they may face challenges with increased stimuli, given that they used only two and six selectable elements, respectively. Regarding the use of other modalities, we also have auditory and tactile options. However, similar to those based on visual overt attention, these modalities generally exhibit poorer performance as the number of available elements increases (e.g., Severens et al. [38], Z. Chen et al. [39], and Séguin et al. [40]). Therefore, it appears that RSVP remains an appropriate visual paradigm for patients without oculomotor control who wish to operate a speller.

### 4.2. ERP Waveform

The RSVP paradigm has the peculiarity that all stimuli—targets and non-targets—are presented at the same spatial location. It implies that the user will perceive both target and, unintentionally, non-target stimuli in their central vision. In contrast to paradigms where stimuli are in different spatial locations, this presents a challenge because non-target stimuli also generate a VEP, making it difficult for the classifier to discriminate between target and non-target stimuli [14]. Hence, in this study, we placed particular emphasis on the amplitude difference as it indicates the variation between target and non-target signals, which is closely related to system performance [9]. Specifically, our results regarding the amplitude difference showed that the distinguishing potentials between target and non-target stimuli are N200 and P300. Both potentials have previously been employed in the literature for ERP-BCI applications and have been manipulated—using different stimuli or SOA—to enhance system performance [41,42]. In our study, the differences in the ERP waveform related to amplitude difference were more affected in terms of latency rather amplitude. Specifically, we observed an earlier onset of components associated with the letter condition. This could potentially stem from differing processing speeds for the two stimulus types (letters vs. faces), which are influenced by the complexity of the visual stimuli [43]. However, it is essential to clarify that our paradigm was not designed for theoretical ERP observation. Instead, what we observe in the corresponding plots is the output of potential underlying components [44]. Therefore, interpretations should be approached cautiously.

### 4.3. Subjective Items

The results regarding the subjective ratings of the participants are more conclusive than those related to performance. While the performance results seem to lean towards recommending the GFF condition, the subjective ratings tend to support the GL condition, even showing significant differences in comfort compared with the other conditions (RFF, GFF, and BFF). However, it is worth noting that the GL condition exhibited significantly lower accuracy compared with the GFF condition. Therefore, the decision regarding which type of stimulus to use will depend on user preferences, needs, and the intended goal of using the ERP-BCI. In addition, there were no significant differences in fatigue based on the type of stimulus employed. However, the ANOVA showed significant differences for letter visibility, although these differences disappeared when correcting for multiple comparisons with the Bonferroni method. Without this correction, there would have been significant differences between the GL condition and each of the face conditions (RFF, *p* = 0.016; GFF, *p* = 0.043; BFF, *p* = 0.015). Therefore, it is possible that the previously mentioned comfort differences may be partly due to the visibility difficulty of the letters when the faces are used, as well as other potential factors not addressed in this study, which could be explored in future research.

### 4.4. Limitations and Future Directions

The present study comes with certain limitations that should be considered, either to clarify the impact of the findings or to address them in future research. The potential limitations include (i) the use of a young and healthy target population; (ii) the restriction on the number of selectable items; (iii) the limited usage time; and (iv) the challenge of recommending a specific condition in the present study.

Firstly, as noted in prior research, verifying outcomes in a clinical demographic remains a hurdle in the BCI domain [45]. In this investigation, most participants were young adult university students without documented neurological conditions. This demographic notably contrasts with the typical target groups for ERP-BCI in RSVP, who are usually middle-aged or elderly individuals with significant motor impairments. Thus, caution is warranted when seeking to extrapolate the findings of this study to the clinical population without corresponding assessments. Nonetheless, it is pertinent to acknowledge that not all BCI applications are tailored to patients; some are intended for non-clinical cohorts, and they could derive value from the insights provided here regarding stimulus types [46].

Second, it is true that the number of items that could be selected is quite limited, especially when compared with other proposals based on an ERP-BCI speller [34]. Using a reduced number of stimuli complicates the generalisation of the results to interfaces offering more stimuli. For example, parameters like the target-to-target interval (TTI)—the time interval between target stimuli, which would be greater with more stimuli—can impact system performance [47]. However, even if TTI affects performance, it would need to be demonstrated that this effect varies depending on the type of stimulus used to invalidate the findings. Furthermore, there are already approaches available that allow a large number of selectable elements with the use of a small number of stimuli, such as those based on the T9 or stepwise selection paradigms [48,49].

Third, we tested four different experimental conditions, and we controlled each condition for a relatively brief period (approximately 10–15 min per condition, encompassing the calibration and online stages). This may deviate from the typical usage experienced by users relying on these systems. Time-related factors such as fatigue or comfort may not manifest until the application is used for an extended duration. Therefore, a potential avenue for future research could involve a more comprehensive and prolonged analysis of usability, which could also help address the following limitation.

Fourth, while this study represents progress in the investigation of visual gaze-independent BCIs for patients lacking oculomotor control, providing a specific interface recommendation presents a challenge. This challenge stems from the fact that, while the GFF condition demonstrated superior performance, it was notably less comfortable compared with the GL condition. Consequently, the decision-making process revolves around what to prioritise: optimising the performance of the ERP-BCI or ensuring user comfort. The choice will be contingent upon user preferences and needs and the intended purpose of employing the ERP-BCI. Although our interface has been tested for a speller, the commands to be selected could be modified with the aim of controlling other types of applications, such as a wheelchair or a home automation system [50,51]. As a result, it is advisable for such decisions to be grounded in real-world usage scenarios. Furthermore, it would be advantageous for the system to be adaptable so that it could cater to evolving user preferences. For example, user preferences may fluctuate within the same day due to user fatigue levels or between sessions as users gain experience with the device. Additionally, future proposals could explore ways to equalise the comfort level of the GFF condition with that of the GL condition, all while preserving the positive impact of the green faces.

## 5. Conclusions

The findings of this study have shown that the type of stimulus used in an ERP-BCI under RSVP significantly affects both user performance and comfort during control. Specifically, there is a trend indicating that famous green faces stimuli lead to enhanced performance; however, the use of faces, as opposed to letters, appears to negatively impact user comfort. This suggests that the selection of stimulus type should be guided by user preferences and system requirements. Consequently, this research holds promise for informing the design of future proposals aimed at controlling these systems. These findings could potentially enhance the quality of life for patients with severe motor impairments, as communication stands as one of their most pressing needs [52]. Therefore, any improvements in the usability of these systems, whether in performance or subjective experience, represent valuable objectives in the field of BCI as AT. However, it is also essential that future proposals validate the findings presented here through experimental replications [53]. Furthermore, there should be a focus on developing an evaluation methodology and using easily adaptable software to enable the selection of optimal parameters (e.g., types of visual stimuli, selectable commands, presentation times, and control paradigms) for each user and control scenario. Additionally, the findings concerning the choice of stimulus type may raise new questions for future studies. We propose the following potential avenues for further research. First, these systems could be evaluated in real-world scenarios with potential users, such as patients with ALS. Second, it would be interesting to evaluate the specific factors that can affect user comfort (e.g., workload [54,55] or usability [56]). Third, it might be relevant to combine the findings of this work with the results from prior research (e.g., alternative gaze-independent paradigms [37]). In conclusion, incorporating the insights from this study with those of prior research is crucial for advancing our understanding and application of ERP-BCI systems, ultimately benefiting users, and addressing their unique needs.

## Figures and Tables

**Figure 1 sensors-24-03315-f001:**
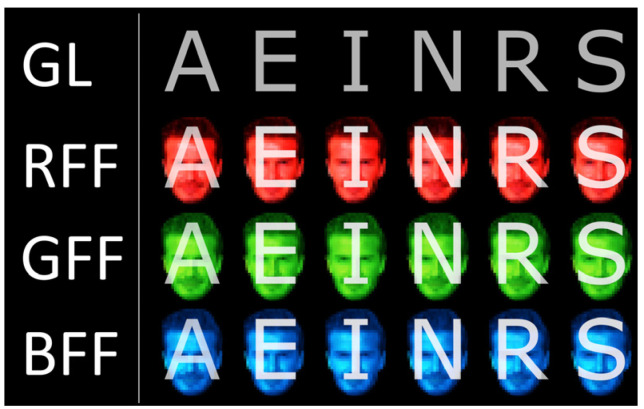
Stimuli used for each condition in the present experiment: grey letters (GL), semi-transparent grey letters with a red famous face (RFF), semi-transparent grey letters with a green famous face (GFF), and semi-transparent grey letters with a blue famous face (BFF). Due to copyright restrictions, David Beckham’s face shown in the figure has been pixelated, unlike in the experiment.

**Figure 2 sensors-24-03315-f002:**
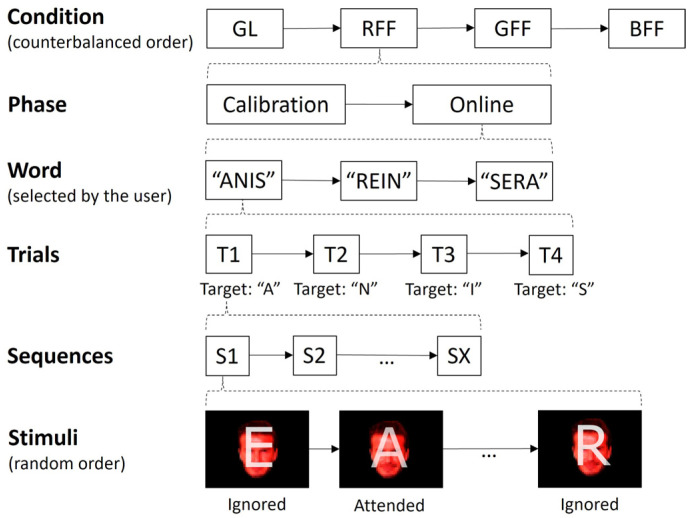
The experimental procedure followed by the participants. The order of the conditions (grey letters, GL; red famous face with letter, RFF; green famous face with letter, GFF; and blue famous face with letter, BFF) was counterbalanced between the participants. Likewise, the order of presentation of the stimuli in each sequence was random, without replacement.

**Figure 3 sensors-24-03315-f003:**
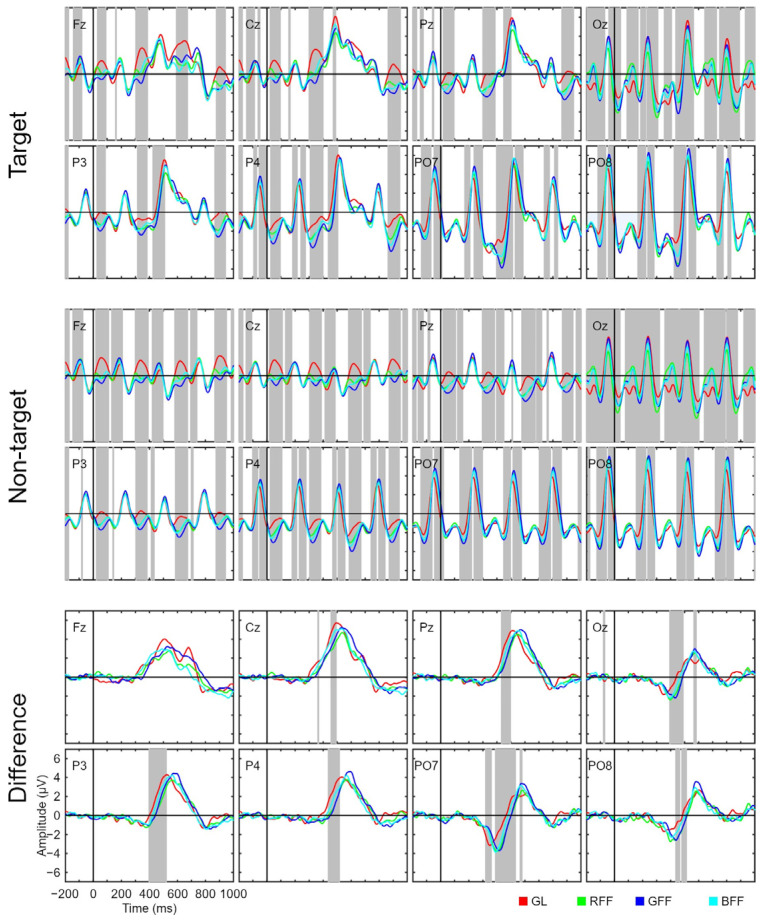
Grand average event-related potential waveforms for the target and non-target stimulus signals, and the amplitude differences between them for all channels used (Fz, Cz, Pz, Oz, P3, P4, PO7, and PO8) and for the four conditions (grey letters, GL; red famous face with letter, RFF; green famous face with letter, GFF; and blue famous face with letter, BFF). Significant intervals are denoted with a grey background on the time axis. The false discovery rate (FDR) correction method was applied.

**Table 1 sensors-24-03315-t001:** The results for each participant in the metrics related to online performance for the variables number of sequences, accuracy (%), and the information transfer rate (ITR, bit/min) for the different experimental conditions (GL, grey letters; RFF, red famous face with letter; GFF, green famous face with letter; and BFF, blue famous face with letter).

Participants	Number of Sequences				Accuracy (%)				ITR (bit/min)			
	GL	RFF	GFF	BFF	GL	RFF	GFF	BFF	GL	RFF	GFF	BFF
P01	7	5	5	3	100	91.67	100	91.67	13.13	14.06	18.38	23.44
P02	4	8	7	4	91.67	100	100	83.33	17.58	11.49	13.13	13.76
P03	5	5	6	3	83.33	66.67	91.67	41.67	11.01	6.35	11.72	2.97
P04	3	6	3	3	91.67	100	100	100	23.44	15.32	30.63	30.63
P05	4	4	4	5	50	75	83.33	91.67	3.769	10.61	13.76	14.06
P06	3	4	5	5	100	83.33	100	100	30.63	13.76	18.38	18.38
P07	4	4	3	4	100	100	100	100	22.98	22.98	30.63	22.98
P08	4	4	6	7	75	91.67	100	100	10.61	17.58	15.32	13.13
P09	5	10	10	7	83.33	83.33	83.33	83.33	11.01	5.504	5.504	7.86
P10	10	9	9	10	66.67	66.67	100	91.67	3.17	3.53	10.21	7.03
P11	3	3	3	4	100	100	100	100	30.63	30.63	30.63	22.98
P12	6	3	4	5	91.67	75	100	75	11.72	14.14	22.98	8.485
P13	3	6	5	5	50	91.67	100	91.67	5.025	11.72	18.38	14.06
P14	7	3	5	3	91.67	91.67	100	91.67	10.05	23.44	18.38	23.44
P15	5	6	8	3	75	91.67	83.33	83.33	8.485	11.72	6.88	18.35
Mean	4.87	5.33	5.53	4.73	83.33 ^1^	87.22	96.11 ^1^	88.33	14.22	14.19	17.66	16.1
SD	1.89	2.12	2.09	1.91	16.39	11.33	6.71	14.53	8.64	6.97	7.88	7.42

^1^ Significant differences (*p* < 0.05) have been found between the averages of these two conditions (GL and GFF) in accuracy.

**Table 2 sensors-24-03315-t002:** The results for each participant in the metrics related to subjective items for the variables fatigue, the visibility of the letters within the visual stimuli, and comfort for the different experimental conditions (GL, grey letters; RFF, red famous face with letter; GFF, green famous face with letter; and BFF, blue famous face with letter).

Participants	Fatigue				Visibility				Comfort			
	GL	RFF	GFF	BFF	GL	RFF	GFF	BFF	GL	RFF	GFF	BFF
P01	3	3	3	4	4	3	3	3	4	3	2	1
P02	2	2	1	1	4	3	2	2	4	1	2	3
P03	2	2	2	2	2	4	3	4	4	2	1	3
P04	2	2	3	2	4	3	2	2	4	2	1	3
P05	2	3	2	4	4	3	3	2	4	3	1	2
P06	4	4	4	3	4	3	3	4	4	1	2	3
P07	1	1	2	2	4	4	3	4	1	4	2	3
P08	3	5	5	4	4	2	0	0	4	3	2	1
P09	1	1	1	2	4	3	4	4	3	1	4	2
P10	4	4	3	5	4	2	3	2	3	2	4	1
P11	3	3	3	3	3	1	4	1	4	2	1	3
P12	2	4	3	4	4	4	1	3	4	1	2	3
P13	3	3	2	2	3	3	4	4	4	2	3	1
P14	1	2	3	3	4	4	3	3	4	1	2	3
P15	4	5	5	4	3	2	4	1	4	2	1	3
Mean	2.47	2.93	2.8	3	3.67	2.93	2.8	2.6	3.67 ^1^	2 ^1^	2 ^1^	2.33 ^1^
SD	1.06	1.28	1.21	1.13	0.62	0.88	1.15	1.3	0.82	0.93	1	0.9

^1^ Significant differences (*p* < 0.05) have been found among the averages of the four conditions, with the comfort reported by participants in GL significantly higher than the other three conditions (RFF, GFF, and GFF).

**Table 3 sensors-24-03315-t003:** Online performance results of Ron-Angevin et al. [35] and the present work; both using a similar brain–computer interface (BCI) based on event-related potentials (ERPs) under rapid serial visual presentation (RSVP).

Study	Condition	Accuracy (%)	ITR (bit/min)
Ron-Angevin et al. [35]	White letters	85.24	13.27
	Famous faces	90.53	17.2
	Pictures	85.99	14.5
Present study	GL	83.3	14.2
	RFF	87.2	14.2
	GFF	96.1	17.7
	BFF	88.3	16.1

Note: ITR, information transfer rate; GL, gray letters; RFF, red famous face with letter; GFF, green famous face with letter; and BFF, blue famous face with letter.

## Data Availability

The data presented in this study are available upon request from the corresponding author.

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
