# Peer review of "Evaluation of Different Types of Stimuli in an Event-Related Potential-Based Brain–Computer Interface Speller under Rapid Serial Visual Presentation"

_sensors, 2024, doi:10.3390/s24113315_

Round 1

Reviewer 1 Report

Comments and Suggestions for Authors

This paper explores the feasibility of a BCI speller based on rapid serial visual presentation (RSVP) in combination with stimuli based on images of faces. Previous studies have explored all of the proposed paradigms, but the authors offer a comprehensive summary and analysis of the combination of these methods with the aim to improve the accuracy and ITR of gaze-independent BCI spellers. The study is properly designed, and the results are adequately presented, with the exception of the "ERP waveform" section and Figure 3, which should be elaborated in more detail.

It is unclear what type of difference the gray intervals of significant differences denote in Figure 3. It seems that they present the differences between conditions (GL, RFF, GFF, BFF). It should be clearly stated for each interval of significant difference between which conditions these differences were observed. Also, it would be interesting to assess the intervals of statistical difference between targets and non-targets for each condition (stimulus type). I suggest that the authors include another figure that shows the intervals of statistical differences between targets and non-targets for each condition. They could color-code the difference intervals in the same manner as the ERP waveforms. This deeper exploratory analysis should provide more detail on how the type of stimuli (face colors and letters) affect not only the ERP morphology but also the differences between targets and non-targets. Generally, visualizations in "ERP waveform" section should be improved and the text should be expanded, describing the statistical differences between conditions and between targets and non-targets within condition in more detail, with potential discussion and interpretations included. 

In the introduction, authors mention that “In essence, any signal that aids in distinguishing the attended stimulus (target) from the unattended ones (non-targets) will be incorporated within the specified time interval (e.g., 0–800 ms after stimulus onset).” I suggest backing up this claim with a recent publication related to the exploration of a tactile ERP paradigm with equiprobable stimuli coupled by a selective attention task (Novičić, M., & Savić, A.M., 2023. Somatosensory event-related potential as an electrophysiological correlate of endogenous spatial tactile attention: prospects for an electrotactile brain-computer interface for sensory training. Brain Sciences, 13(5), p.766)." Moreover, even though the stimulation modality is different, in terms of attention effects and type of components induced, they may find these results interesting for comparison.

Author Response

First and foremost, we would like to express our gratitude for the time and effort the reviewer has dedicated to providing feedback and assisting us in enhancing the quality of the article. Enclosed within the attached document are the detailed responses and modifications made to the manuscript. This response document encompasses comments from both reviewers.

Reviewer 2 Report

Comments and Suggestions for Authors

The presented manuscript aimed at evaluating whether the colour of faces used as visual stimuli influences ERP-BCI performance under Rapid serial visual presentation, through an BCI-based approach. The work is well organized, and the results appear to be significant. However, there are different consistent issues to be addressed before considering it for publication. More specifically:

1. The Authors vaguely introduced the concept of BCI. In this regard, the Introduction section should be updated by adding more specific previous contributions from the scientific community to the central topic of the presented research.

2. Within the Material and Methods section, it is not described how the EEG signal was processed in order to detect and correct signal artifacts.

3. The performed statistical analysis should be better described.

4. Along the entire manuscript, it is not clear the exact added value provided by the present research with the respect to the state of the art. In this regard, the Authors should better contextualize their work, by citing the most transversal and updated research works.

5. The Discussion section is lacking by a consistent description of the practical outcome and future employments of the proposed approach.

6. Within the Discussion section it is not clear how the different stimuli (not only visual) can impact on the brain patterns. In this regard, different recent works in literature could be considered for further describing this aspect (such as G. Borghini et al., "Stress Assessment by Combining Neurophysiological Signals and Radio Communications of Air Traffic Controllers," 2020 42nd Annual International Conference of the IEEE Engineering in Medicine & Biology Society (EMBC), Montreal, QC, Canada, 2020, pp. 851-854, doi: 10.1109/EMBC44109.2020.9175958; Impact of Stimulus Features on the Performance of a Gaze-Independent Brain-Computer Interface Based on Covert Spatial Attention Shifts).

Comments on the Quality of English Language

There are different repetitions in the text, I would suggest to go through the manuscript and enrich the terminology.

Author Response

(The authors gave the same response as above.)

Round 2

Reviewer 1 Report

Comments and Suggestions for Authors

The authors have successfully addressed my comments.